# Sources of Hallucination by Large Language Models on Inference Tasks

**Nick McKenna**[†][*]    **Tianyi Li**[†][*]
**Liang Cheng**[†]    **Mohammad Javad Hosseini**[‡]    **Mark Johnson**[§]    **Mark Steedman**[†]
[†]University of Edinburgh    [‡]Google Research    [§]Macquarie University
{nick.mckenna, tianyi.li}@ed.ac.uk

## Abstract

Large Language Models (LLMs) are claimed to be capable of Natural Language Inference (NLI), necessary for applied tasks like question answering and summarization. We present a series of behavioral studies on several LLM families (LLaMA, GPT-3.5, and PaLM) which probe their behavior using controlled experiments. We establish two biases originating from pretraining which predict much of their behavior, and show that these are major sources of hallucination in generative LLMs. First, memorization at the level of sentences: we show that, regardless of the premise, models falsely label NLI test samples as entailing when the hypothesis is attested in training data, and that entities are used as "indices" to access the memorized data. Second, statistical patterns of usage learned at the level of corpora: we further show a similar effect when the premise predicate is less frequent than that of the hypothesis in the training data, a bias following from previous studies. We demonstrate that LLMs perform significantly worse on NLI test samples which do not conform to these biases than those which do, and we offer these as valuable controls for future LLM evaluation.[1]

## 1 Introduction

Large Language Models (LLMs) such as LLaMA, GPT-3/4, PaLM, and more (Touvron et al., 2023; Brown et al., 2020; Chowdhery et al., 2022), have been trusted by many to perform language understanding in downstream tasks such as summarization, question answering, and fact verification, among others (Zhang et al., 2023). However, due to the large-scale nature of LLM training on vast, often proprietary data, and the inherent opacity of LLM parameters, it is difficult to explain their behavior when answering user queries and the corresponding risks in terms of bias and robustness. In particular, one LLM behavior poses a significant challenge: "hallucination," the phenomenon in which LLMs provide information which is incorrect or inappropriate, presented as fact.

This paper investigates two biases driving LLM performance in natural language inference, sometimes called *textual entailment*. This is a basic component of language understanding which is critical in applied tasks, and we offer these two biases as explanations of general false positive hallucination in everyday use. We examine broader NLI, and especially *directional entailments*, which hold in one direction, but not both. For example, DEFEAT *entails* PLAY but PLAY *does not entail* DEFEAT. Inferring directional entailment is more difficult than that of symmetric paraphrase, so it more deeply probes understanding.

Our approach is a behavioral study of prompted LLM decision-making. We alter existing NLI datasets in targeted ways while measuring how predictions change, across several major LLM families (LLaMA, GPT-3.5, and PaLM). We demonstrate two sources of LLM performance on the NLI task, which we offer as explanations of general false positive hallucination: (1) LLM bias toward affirming entailment when the hypothesis may be attested in the training text, including reliance on named entity identifiers; and (2) a corpus-frequency bias, affirming entailment when the premise is less frequent than the hypothesis.

We establish that these biases originate from the LLM pretraining objective, in which statistical modeling of the natural distribution of human-generated text leads to (at the level of sentences) memorizing individual statements, and (at the level of corpora) learning typical patterns of usage. Though they are superficially performant, our experiments show that even powerful LLMs still use unsatisfactory tools instead of robust reasoning.

---

[*]Equal contribution.

[1]Code and LLM outputs (LLaMA and GPT-3.5) are available at https://github.com/Teddy-Li/LLM-NLI-Analysis.

We present three contributions, the demonstrations of both factors and an analysis of their impact:

(1) In a prompting scenario, LLMs respond to entailment samples according to an *attestation bias*, affirming entailments more readily if the hypothesis is attested by the pretraining text. We find that LLaMA-65B, GPT-3.5, and PaLM-540B are respectively 1.9, 2.2, and 2.0 times more likely to wrongly predict `Entail` when the model already asserts the hypothesis is attested, compared to when not attested. Further, LLMs recall from their propositional memory using named entities as identifying "indices," even though they are irrelevant to the logic of the predicate inference task.

(2) LLMs also rely on a simple corpus-statistic bias using relative term-frequencies, especially when propositional memory is not available. The three LLMs are 1.6, 1.8 and 2.0 times more likely to wrongly affirm entailments if the premise has lower term frequency than the hypothesis, than when not.

(3) For the NLI test samples consistent with these factors, LLM scores are misleadingly high; for NLI samples adversarial to them, LLM performance is severely degraded. We show that when labels go against the *attestation bias*, LLMs can be poor or even near-random classifiers; for the *relative frequency bias*, we similarly show a substantial performance decrease across all LLMs.

## 2 Related Work

Addressing task robustness, Poliak et al. (2018) found a range of NLI datasets contain artifacts which are learned by supervised models trained on only sample hypotheses. In our work we design a similar hypothesis-only test with LLMs, but we use it to probe model memory without any training. By conditioning on the attestation of hypotheses, we show that LLMs are inherently sensitive to attestation, separate from the statistical idiosyncrasies of NLI datasets.[2]

Additionally, Talman and Chatzikyriakidis (2019) report generalization failure among many models supervised for NLI — models fail to generalize between NLI datasets, even if the task is formatted the same. On smaller Language Models such as RoBERTa (Liu et al., 2019; 355M params), Li et al. (2022) also observed a reliance on dataset

artifacts when performing directional NLI on predicates. We now study the behavior of much larger LMs, which have demonstrated more robust performance across NLP tasks.

Recent work has also explored LLM memorization and generalization. Carlini et al. (2023) establish that LLMs are able to memorize more data than small LMs, whereas Tirumala et al. (2022) further demonstrate that LLMs pay special attention early in training to numbers and nouns, which act as unique identifiers for individual training sentences. We also show that memories used in language inference are tied to specific named entities. And while Weller et al. (2023) and Kandpal et al. (2022) find that entity frequency in training data is correlated with performance in factual recall about them, we find that entity frequency is *anti*-correlated with hypothetical generalization performance (§6).

Bubeck et al. (2023) argue that GPT-4 understands language "beyond memorization". We do not disprove generalization, but we show that GPT-4 shows the same hallucinations in Appendix F.

## 3 Experimental Design

We design behavioral experiments on LLMs by modifying NLI datasets with rigorous controls. We observe large behavior changes across three major LLM families due to propositional-memory effects in §5 and §6, and corpus frequency in §7. Finally, we show the impact on real performance in §8.

### 3.1 Two Biases in Inference Predictions

We claim that the pretraining objective to fit the distribution of natural text leads to biases in LLM generations. We explore two such biases.

**The Attestation Bias ($\Lambda$)** is the over-reliance of an LLM on its propositional memory about a query statement. We claim that when a statement is likely to be attested in some way by an LLM's training data, it is more likely to affirm it as a conclusion in NLI tasks, regardless of any premise it is presented with. We measure the attestedness of a sample by prompting the LLM to ask if the hypothesis in question is true, false, or unknown.[3] Attestation predictions are denoted by $\Lambda$.

A biased model will appear to perform well on dataset samples with entailment labels that happen to align with the bias. Table 1 shows examples from the Levy/Holt dev set.

---

[2] We speculate that a similar attestation effect could even be present in the supervised models studied in Poliak et al. (2018), which could contribute to those models' performance. We leave the investigation of this to future work.

[3] Alternatively, LLM perplexity for a statement could be used; however, this is not always available, e.g. with GPT-3.

| Dev Sample Query: [premise] ⇒ [hypothesis] | Dataset Label | Bias Prediction |
|---|---|---|
| Geysers are common to New Zealand ⇒ Geysers are found in New Zealand | `Entail` | $\Lambda$ = hypothesis `Attested` |
| Geysers are found in New Zealand ⇒ Geysers are common to New Zealand | `No-Entail` | $\Lambda$ = hypothesis `Not-Attested` |
| Whiskey consists chiefly of alcohol ⇒ Whiskey contains alcohol | `Entail` | $\Phi = f(\textit{consists chiefly of}) < f(\textit{contains})$ |
| Whiskey contains alcohol ⇒ Whiskey consists chiefly of alcohol | `No-Entail` | $\Phi = f(\textit{contains}) > f(\textit{consists chiefly of})$ |

Table 1: Two pairs of samples are consistent with a respective bias. Model predictions made on the basis of the bias will appear to predict the direction of entailment for each sample. $f(\cdot)$ maps a term to its corpus frequency.

As discussed in §2, we draw inspiration from the *hypothesis-only baseline* (Poliak et al., 2018), but our test only queries model memory about the hypothesis without any training. We describe prompt generation in detail in §4.2, with an example in appendix Table 13.

Dasgupta et al. (2022) show a similar effect in LLMs on abstract reasoning tests, related to sentential content, and equate it to human tendencies. In contrast, we examine the risks of propositional memory on more realistic inference tasks.

**The Relative Frequency Bias ($\Phi$)**  is the use by LLMs of a simple rule for deciding entailment, calculable from corpus statistics. Entailment is affirmed if, ignoring named entities, the eventuality in premise $P$ is less frequent in training than that in hypothesis $H$.

This bias is reflected in natural text: it is known that nouns follow a trend of becoming more specific as corpus-frequency decreases (Rosch et al., 1976; Caraballo and Charniak, 1999) and the same is observed for predicates (McKenna et al., 2023). Since entailments may carry from a specific term to a more general one, e.g. SPRINT *entails* RUN, relative frequency can often indicate direction of entailment. However, this is an artifact of natural text and has no direct relationship with meaning.

Test samples are labeled for agreement with this bias separately from models, with examples shown in Table 1. Since LLM pre-train corpora are impractically large or proprietary, we instead use Google N-grams[4] as a proxy of the natural distribution of text, and thus the distributions of these corpora. We average frequencies between the years 1950-2019, and compare between $P$ and $H$. To acquire the generic eventualities in each sample, we exclude any extracted entities and lemmatize predicate phrases; further, we reduce the effect of noise and sparsity by requiring a wide margin of difference between $P$ and $H$ frequency estimates. Frequency decisions are denoted by $\Phi$.

[4] https://books.google.com/ngrams

### 3.2 Datasets

**Levy/Holt**  consists of premise-hypothesis pairs, with a task formatted: "Given [premise $P$], is it true that [hypothesis $H$]?" (Levy and Dagan, 2016; Holt, 2019). Each $P$- and $H$-statement has the property of containing one predicate with two entity arguments, (where the same entities appear in both $P$ and $H$) as shown in Table 2. This targeted dataset is ideal for precisely measuring model understanding of predicates, because entailment between statements is decidable purely on the basis of the predicates and their attributes. We study the challenging directional subset, where entailments hold in one direction but *not* both.

**RTE-1**  is one of the original and most difficult tests of NLI (Dagan et al., 2006). It is not purely directional on the basis of predicates or consistently structured like Levy/Holt, so we leave it out of the behavioral experiments. However, it is a widely understood dataset, and we use it to demonstrate the impact of the two biases in general NLI in §8.

**Exclusions**  are made of NLI datasets relating to knowledge of the world, since we aim to test LLMs on their capability to reason purely about the semantics of natural language predicates without relying on memorized facts. We explicitly avoid datasets such as MMLU (Hendrycks et al., 2021), Natural Questions (Kwiatkowski et al., 2019), OpenBookQA (Mihaylov et al., 2018) etc.

### 3.3 Dataset Transformations

**The Standard Inference Task ($I$)**  is on original NLI datasets, in which entailment is determinable by using general language inference on sentences. In Levy/Holt, it is determinable just by predicates.

We define three dataset transformations to study the change in model responses as targeted information is removed. These are the randomized premise predicate setting $I_{RandPrem}$, and two argument transformations: generic arguments $I^{GenArg}$, and type-constrained randomized arguments $I^{RandArg}$.

| Task | Label | Dev Sample Query: [premise] ⇒ [hypothesis] |
|---|---|---|
| $I$ | Entail | George Bush was the Governor of Texas ⇒ George Bush is a politician from Texas |
| $I_{RandPrem}$ | No-Entail | George Bush resided in Texas ⇒ George Bush is a politician from Texas |

Table 2: From the original dataset task ($I$) we derive the Random Premise task ($I_{RandPrem}$), respecting type-constraints. A random premise is highly unlikely to entail the hypothesis, so samples are relabeled No-Entail.

| Task | Label | Dev Sample Query: [premise] ⇒ [hypothesis] |
|---|---|---|
| $I$ | Entail | India exports tons of rice ⇒ India exports rice |
| $I^{GenArg}$ | Entail | location X exports tons of food Y ⇒ location X exports food Y |
| $I^{RandArg\downarrow}$ | Entail | Sloterdijk exports tons of oatmeal cookies ⇒ Sloterdijk exports oatmeal cookies |
| $I^{RandArg\uparrow}$ | Entail | Helsinki exports tons of Granny Smith ⇒ Helsinki exports Granny Smith |

Table 3: An original dev sample ($I$) is transformed by insertion of entity types ($I^{GenArg}$); by real entities sampled from the 5% least frequent in NewsCrawl ($I^{RandArg\downarrow}$); and also from the 5% most frequent ($I^{RandArg\uparrow}$).

Transformations involve first identifying the types of entities in statements, in order to constrain entity or predicate replacements. We type each entity with one of the 48 FIGER types (Ling and Weld, 2012), such as "person," "location," etc. First, an entity linker (Nguyen et al., 2014) identifies the Freebase ID (Bollacker et al., 2008) for an entity, from which we then obtain its FIGER type; we assign a default type "thing" in failure cases.

**The Random Premise Task ($I_{RandPrem}$)** replaces the original premise predicate with a random predicate, while maintaining the same entity arguments. This manipulation produces a dataset in which all samples are labeled No-Entail, since two randomly paired predicates are very unlikely to be related by entailment. Thus, positive decisions by the model are false positive hallucinations.[5]

To maintain naturalness and grammaticality, we constrain a new predicate to have argument slots of the same types as the original premise. For example, "[medicine] is indicated for patients with [disease]" is swapped for "[medicine] does not cure [disease]". We source candidates from dev set premises satisfying the target type-constraints, and sample uniform randomly. We map the original entities to their respective slots in the new premise. Examples are shown in Table 2. $I_{RandPrem}$ is a good test of model reliance on propositional memory, since we prevent entailments while maintaining the attestedness of conclusions (hypotheses).

**The Generic Argument Task ($I^{GenArg}$)** replaces original entities with unique FIGER-typed

identifiers, e.g. "location X" and "food Y." By masking the identities of entities, this test is designed to remove entity information while maintaining the same entailment label, as a baseline control setting. We append unique identifiers (e.g. "X," "Y") to allow tracking of entity slots across the premise and the hypothesis.

**The Random Argument Task ($I^{RandArg}$)** replaces original entities with other real, random entities of the same FIGER-type. Like $I^{GenArg}$, this test is designed to create novel strings by modifying statements without changing entailment labels. But now we test model sensitivity to added extraneous information. Examples are shown in Table 3.

We use entity type constraints here to ensure polysemous predicates maintain the same sense. For example, a different sense of *run* is used in "[person] runs [organization]" vs. "[person] runs [software]", but between different entities of the same type, the same senses are used, so the exact entity IDs do not affect entailment labels (Yarowsky, 1993). We source new entities from NewsCrawl (Barrault et al., 2019), a decade-long span of multi-source news text, in which entities are typed as above. We sample new entities uniform randomly from the 5% least common entities in NewsCrawl ($I^{RandArg\downarrow}$), and the 5% most common ($I^{RandArg\uparrow}$). We insert the sampled entities while preserving the rest of each statement.

## 4 Querying Models with Prompts

### 4.1 Models

**LLaMA** is a recent LLM model family which rivals or surpasses GPT-3 performance while being open to scientific study. A range of model sizes

---

[5]We manually inspected the generated random premise entries for the Levy/Holt dataset to verify this: we found 86.6% of entries are successfully non-entailing, 3.8% undecided cases, and only 9.6% are unintended true entailments.

are provided, and we test the largest **LLaMA-65B** model. LLaMA is not fine-tuned. In preliminary experiments on the Levy/Holt dataset, we found two popular fine-tuned LLaMA variants, Alpaca (Taori et al., 2023) and Vicuna (Chiang et al., 2023), perform similarly to LLaMA base models and underperform LLaMA-65B, so we leave them out of further experiments.

**GPT-3 Series** models are closed to deep scientific review (Brown et al., 2020), though they are a widely-used comparison for their performance, and have been reasonably well-studied. We evaluate on **text-davinci-003 (GPT-3.5)**, as it is the largest, and has undergone instruction- and RLHF-finetuning, enabling interesting comparisons.

**PaLM** is larger than GPT-3, which often claims state-of-the-art on evaluation datasets. We use the largest **PaLM-540B** base model, which is also only pretrained, so it serves as a further comparison point to LLaMA.

Later GPT models (like text-davinci-003 in our experiments) have been pre-trained and fine-tuned, while base LLaMA and PaLM have only undergone pre-training, so their contrast indicates what stage of training is responsible for the phenomena we study. Our aim is not to judge which LLM is superior, but to show the common sources of hallucination they share.

We also omit models superseded in performance by LLaMA (e.g. OPT, GPT-J, etc.), as well as products that are closed to scientific review (e.g. GPT-4, Bard, etc.)[6].

### 4.2 Prompt Design and Evaluation

**Formatting** of test samples is done by inserting the premise and hypothesis into a prompt template, which is used to query the model in natural language. Following this, we append a three-way answer choice: A) Entailment, B) Neutral, C) Contradiction, following the typical format in NLI (Bowman et al., 2015).

**Selection** of the prompt template used in test is decided by the highest AUC obtained on the respective dev set. We try 8 promising templates including 5 from Schmitt and Schütze (2021), also used in other NLI work[7] (Webson and Pavlick, 2022).

---

[6]We include an analysis of GPT-4 in Appendix F.

[7]See Appendix A for details on prompt template selection.

Ideally, an LLM with advanced language understanding ability could perform inference in zero-shot without annotated examples, which would raise confidence that this faculty is ready for downstream tasks. To this end, we examine each LLM in zero-shot (detailed in Appendix A), but they exhibit severely degraded, even near-random performance.

We turn to few-shot, and hand-annotate a minimal 4 examples in the style of the template, with added explanations about why the given answer is correct for each example. These examples are prepended before the query (see Appendix A for an example). Our goal is to study model behavior as conditions change, not to maximize the score on any particular dataset. Therefore, we use a minimal 4-example setup, which we find is capable of evoking positive responses from all three LLMs on each dev set, across most templates.

**Scoring** is done by converting choice A into `Entail` and collapsing both B and C choices into `No-Entail` to align with Levy/Holt and RTE-1 annotation. For behavioral experiments in §5, §6, and §7, we score the model solely based on its textual response. All models successfully choose one of A/B/C on all dev questions, showing compatibility with the QA format.

For the analysis in §8 which measures model performance across confidence thresholds, we convert the letter choice to a probability with the mapping:

$$S_{\text{ent}} = 0.5 + 0.5 * \mathbb{I}[\text{tok} = \mathbf{A}] * S_{\text{tok}}$$
$$- 0.5 * \mathbb{I}[\text{tok} \in \{\mathbf{B}, \mathbf{C}\}] * S_{\text{tok}}$$

Where $\mathbb{I}$ is the indicator function, and $S_{ent}$ estimates the probability of `Entail` from a textual output ($0 \leq S_{\text{ent}} \leq 1$) with token probability $S_{tok}$. The linear transformation preserves the ordering of model confidences, which is sufficient for calculating a precision-recall curve.

## 5 Experiment 1: Attestation Bias

We begin our experiments by assessing LLMs' reliance on their propositional memory of training text by conditioning each model's entailment task predictions $I$ on its own predictions of attestation $\Lambda$. We do this by comparing estimated probabilities of predicting `Entail` conditioned on whether the hypothesis is predicted `Attested` or not.

Further, we test a setting which controls for the possibility that original Levy/Holt entailments may coincidentally refer to attested facts, which could

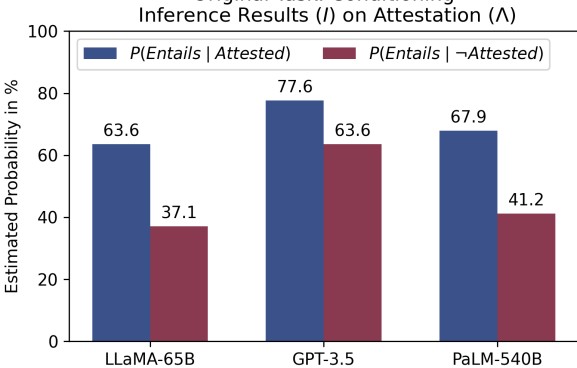

Figure 1: Exp-1. Estimated probability of predicting `Entail` for **original** entries in Levy/Holt, conditioned on LLMs' attestation of hypotheses (Λ). This setting is intuitive but may be subject to spurious correlations.

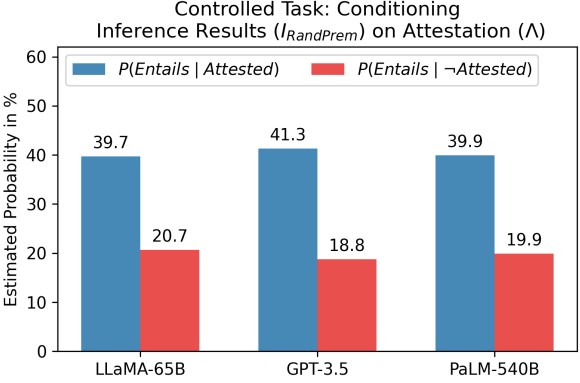

Figure 2: Exp-1. Estimated probability of predicting `Entail` for **Random-Premise** entries in Levy/Holt, conditioned on LLMs' attestation of hypotheses (Λ). Now, predicting `Entail` is false positive hallucination (lower is better). Models are sensitive to attestation, and hallucinate more when the hypothesis is attested.

lead to spurious correlation between inference and attestation scores without clearly demonstrating use of memory versus true entailment. This controlled setting is the random premise task $I_{RandPrem}$, which converts entailments into non-entailments without altering the hypothesis. An ideal model capable of drawing inferences from information in context should detect that in the $I_{RandPrem}$ task it is no longer possible to infer the hypothesis based on the premise (even if the hypothesis is itself attested in training), and never predict `Entail`. Thus, in $I_{RandPrem}$, all `Entail` predictions are assumed to be false positive hallucinations.

## 5.1 Results

With $I$, $I_{RandPrem}$ and Λ predictions acquired as described in §3.1, we present the conditional probabilities in Figures 1 and 2. It is clear that a model's

memory about the hypothesis plays a part in its predictions of the hypothesis given a premise, either related or random.

For $I$, we observe significantly higher probability of predicting `Entail` when the hypothesis is `Attested`. In the random premise task $I_{RandPrem}$, this trend continues. LLaMA, GPT-3.5, and PaLM, respectively, show a 1.9x, 2.2x, and 2.0x higher chance of falsely predicting that a random premise `Entails` the hypothesis if it already predicts the hypothesis is `Attested`. This false positive hallucination and its impact on NLI performance is investigated further in §8.

This behavior is observed across model families (LLaMA, GPT, and PaLM), establishing that it is due to pretraining rather than Instruction-tuning or RLHF, since LLaMA and PaLM have only undergone pretraining. This behavior is undesirable, because model predictions on NLI tasks should be based solely on general language understanding, not prior knowledge. We may conclude that memory of training data is a significant contributor in LLM inference, and may be an important source of hallucination.

## 5.2 Implications for Real Applications

Using prior knowledge as part of language inference has bad implications for the use of LLMs in real applications. We offer an example scenario of a question-answering task where user questions are answered from a Knowledge Base (KB). In typical formulations of this task, if a statement in the KB (premise) entails a user query (hypothesis), the premise may be formulated into an answer. Consider a KB such as a legal document or HR rulebook. Assume that the text is prepended to the user query and presented to the LLM, as in other works (Srinivasan et al., 2022). Given our findings, we might observe the LLM hallucinating answers to questions using information which is not presented in the KB, but may have been read by the LLM in text from other sources during pretraining. These answers could be illogical, contradictory, and could misrepresent the views of the KB, or other harms. Such poor use of in-context learning has already been observed in specific domains like medicine (Jimenez Gutierrez et al., 2022).

In general, this is a risk for LLMs which (a) are deployed for tasks like QA by feeding novel text (e.g. a legal document) in-context as part of the user query, and (b) are trained on datasets which are

| Model | Task | Levy/Holt (Directional) | | |
|---|---|---|---|---|
| | | Precision | Recall | $\Delta$-*Recall* |
| LLaMA | $I$ | 67.0 | **68.4** | *0* |
| | $I^{GenArg}$ | 69.0 | 66.9 | *-1.5* |
| | $I^{RandArg\downarrow}$ | 64.0 | 63.8 | *-4.6* |
| | $I^{RandArg\uparrow}$ | 67.2 | 53.7 | *-14.7* |
| GPT-3.5 | $I$ | 62.4 | **92.3** | *0* |
| | $I^{GenArg}$ | 65.1 | 75.7 | *-16.6* |
| | $I^{RandArg\downarrow}$ | 65.5 | 66.5 | *-25.8* |
| | $I^{RandArg\uparrow}$ | 68.8 | 55.3 | *-37.0* |
| PaLM | $I$ | 72.8 | **76.2** | *0* |
| | $I^{GenArg}$ | 79.8 | 50.8 | *-25.4* |
| | $I^{RandArg\downarrow}$ | 69.5 | 58.7 | *-17.5* |
| | $I^{RandArg\uparrow}$ | 70.8 | 52.4 | *-23.8* |

Table 4: Exp-2. Scoring model outputs in different argument-replacement tasks. We indicate the **highest** and lowest recall score across replacement settings. Recall decreases sharply across settings in all models.

private or otherwise infeasibly large to read manually, containing many facts and human opinions unknowable to both the user and modeler.

# 6 Experiment 2: Entities are Indices to Memory

In §5, we have established that propositional memory explains a significant portion of false positives in LLM inference predictions. In this section, we continue by showing the importance of named entities in the process of LLMs' memory recall.

As described in §3.3, we manipulate the entities with the $I^{GenArg}$ generic argument replacement, and two random entity replacements, one with infrequent-entities $I^{RandArg\downarrow}$ and one with frequent-entities $I^{RandArg\uparrow}$ (examples in Table 3).

By replacing arguments constrained by type, entailment labels are maintained; however, new samples should contain novel strings not attested in pretrain corpora. We expect that an ideal, generalizing model would maintain its predictions across all conditions; a flawed model utilizing the *attestation bias* would predict fewer `Entail`, since entities no longer identify these statements in training.

## 6.1 Results

We report results across conditions in Table 4. We observe two phenomena across all three models, aligning with the above conjecture of "flaws."

First, we observe that all models' behavior significantly changes in the same way when original entities are replaced by either entity types or random real entities. Despite similar (or marginally increas-

ing) precision across conditions, recall degrades sharply from original entities ($I$) (GPT-3.5 @92.3) to random frequent entities ($I^{RandArg\uparrow}$) (GPT-3.5 @55.3). Generic-argument $I^{GenArg}$ performance also degrades in this way, showing that this is not a matter of poorly selected real entities, but rather a loss of information from the original dataset which models were using to answer questions.

Second, across the 3 models, we observe a significant difference in recall between the two real entity conditions $I^{RandArg\downarrow}$ and $I^{RandArg\uparrow}$, which are both composed of unattested statements, but involve entities that differ in typical corpus frequency. Infrequent entities ($I^{RandArg\downarrow}$) yield better generalization and a higher recall (GPT-3.5 @66.5) than frequent entities ($I^{RandArg\uparrow}$) (GPT-3.5 @55.3).

These findings corroborate those from §5, that LLMs use memory as part of language inference, and additionally show that these memories are recalled using named entities acting as indices. These experiments demonstrate that too much prior exposure to an entity may impede model generalization when that entity is discussed in novel inferences: the more a model has read about an entity during pretraining, the less capable it is of drawing novel natural language inferences involving it. This is the case even though the inferences do not require detailed knowledge of the entity.

Like §5, the effect is consistent across models, indicating LLM pretraining is responsible.

We show similar results on RTE-1 in Appendix B. Further, instructing LLMs to ignore propositional memory in Appendix C shows little change.

# 7 Experiment 3: Relative Frequency Bias

We continue the conditioning experiments from §5, now exploring the relative frequency bias. Sample labels for this bias are denoted by the model-agnostic $\Phi$ as described in §3.1. $\Phi$ labels the conformance of sample predicates to the bias: $\Phi_<$ means $P$ is less corpus-frequent than $H$ by a margin (positive class), $\Phi_>$ means $P$ more frequent than $H$ by the margin (negative class). To control for differences between datasets, the margin is set so that 1/3 of samples are classed as "roughly equal" ($\Phi_\approx$), which we discard.

Following the observations in §6, we further apply a generic-argument transformation to control for attestation, yielding $I^{GenArg}_{RandPrem}$. With the entities masked, models cannot recall propositional memory for this task: by re-calculating the $\Lambda$ mea-

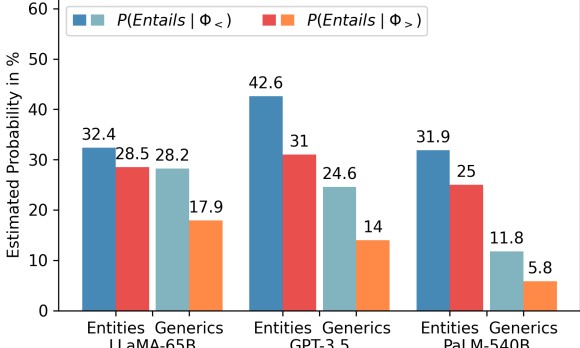

Figure 3: Exp-3. Estimated probability of predicting `Entail` for **random-premise** Levy/Holt conditioned on relative frequencies ($\Phi$), with original ($I_{RandPrem}$) or generic ($I_{RandPrem}^{GenArg}$) entities. Predicting `Entail` is false positive hallucination (lower is better). Models hallucinate more often when test samples conform to the relative frequency bias ($\Phi_<$) than when not ($\Phi_>$).

sure with generic arguments, only 2 hypotheses are still predicted as `Attested` by GPT-3.5, whereas for LLaMA and PaLM, the numbers are also only 6.2% and 3.9%. Additionally, as with $I_{RandPrem}$, here the entailment label of each sample remains `No-Entail`, so any `Entail` prediction is false positive hallucination.

### 7.1 Results

We estimate the probabilities of models predicting `Entail` conditioned on the frequency label $\Phi$, between $I_{RandPrem}$ and $I_{RandPrem}^{GenArg}$ settings, and present the results in Figure 3. We observe a clear and consistent rise of hallucination when samples conform to the bias. Namely, in case of $\Phi_<$, models are more likely to predict `Entail`, even though no semantic relation exists between $P$ and $H$.

Between the two settings, with $I_{RandPrem}$, when entities are available, this effect is moderate. On the other hand, with $I_{RandPrem}^{GenArg}$ when entity-based memory is blocked, we observe a decrease in the overall level of hallucination, but the separation between $\Phi_<$ and $\Phi_>$ becomes more drastic, to 1.6x, 1.8x and 2.0x for LLaMA, GPT-3.5 and PaLM respectively. This indicates a tension between $\Lambda$ and $\Phi$: propositional memory may be used when available, and if not, the predicate pairing may be attended to more closely. Again, the $\Phi$ effect is observed across the three model families, revealing its root in the large-scale pre-training process, rather than model peculiarities or fine-tuning.

## 8 Impact of Bias on Performance

We have demonstrated two sources of hallucination by LLMs on inference tasks. We now assess their impact on model performance to quantify their risk.

We compare LLMs' performance between NLI subsets that are *consistent* or *adversarial* to each bias. A sample $P \vDash H$? is *consistent* with a bias when the prediction by the bias **agrees with** the gold entailment label; conversely, it is *adversarial* to a bias when the prediction by the bias **disagrees with** the label.

For example, "Google bought YouTube $\vDash$ Google owns YouTube" is *consistent* with the attestation bias of every model, because the conclusion *Google owns YouTube* is attested in every LLM's training data, and the sample label is `Entail`; "Apple owns Samsung $\nvDash$ Apple bought Samsung" is also *consistent*, because its conclusion is not attested and the sample label is `No-Entail`. The reverses of these two samples are *adversarial*, since their respective attestedness (unchanged) does not agree with the entailment labels (now flipped). For each subset, there is substantial representation in both Levy/Holt and RTE-1 (see appendix Table 9).

While earlier experiments inspected model textual responses to characterize behavior change, we now use area under the precision-recall curve (AUC) to summarize model performance over a tunable confidence threshold (scoring described in §4.2), which is better for measuring practical discriminative power. Following Li et al. (2022), we re-scale AUC values to normalize over the label distribution, yielding $AUC_{norm}$ values that assign random classifiers 0% and perfect classifiers 100%.

We report results in Table 5. Under the standard inference task $I$, the performance drop from $\Lambda_{\text{CONSISTENT}}$ to $\Lambda_{\text{ADVERSARIAL}}$ is severe for all 3 LLMs: they deteriorate from very good classifiers to poor or even near-random ones.[8] This fragility from the *attestation bias* can be alleviated by masking entities with type-identifiers (condition $I^{GenArg}$), which reduces the performance drop.

On the other hand, with the generic arguments in $I^{GenArg}$, LLMs are forced to focus on the predicates in each proposition. As a result, the impact of the *relative frequency bias* is intensified. From the standard inference task $I$ to $I^{GenArg}$, the average performance drop from the *cons.* to *adv.*

---

[8]We note $\Lambda$ predictions could possibly be influenced by model-specific idiosyncrasies in prompt format. We provide an analysis in Appendix E, where we find no significant effect.

| Model | Task | Levy/Holt | | | | | | RTE-1 | | | | | |
|---|---|---|---|---|---|---|---|---|---|---|---|---|---|
| | | Attestation ($\Lambda$) | | | Rel. Frequency ($\Phi$) | | | Attestation ($\Lambda$) | | | Rel. Frequency ($\Phi$) | | |
| | | *cons.* | *adv.* | *diff.* | *cons.* | *adv.* | *diff.* | *cons.* | *adv.* | *diff.* | *cons.* | *adv.* | *diff.* |
| LLaMA | $I$ | 65.5 | 8.1 | *-57.4* | 42.1 | 32.3 | *-9.8* | 62.1 | 37.4 | *-24.7* | 55.5 | 51.7 | *-3.8* |
| GPT-3.5 | $I$ | 85.0 | 10.8 | *-74.2* | 53.5 | 43.2 | *-10.3* | 84.6 | 47.5 | *-37.1* | 77.6 | 43.4 | *-34.2* |
| PaLM | $I$ | 79.1 | 31.5 | *-47.6* | 63.3 | 53.0 | *-10.3* | 87.1 | 83.4 | *-3.7* | 87.5 | 81.0 | *-6.5* |
| LLaMA | $I^{GenArg}$ | 52.1 | 34.4 | *-17.7* | 55.3 | 34.9 | *-20.4* | 59.2 | 30.4 | *-28.8* | 51.7 | 39.4 | *-12.3* |
| GPT-3.5 | $I^{GenArg}$ | 67.1 | 18.8 | *-48.3* | 50.4 | 35.0 | *-15.4* | 80.1 | 56.4 | *-23.7* | 79.6 | 49.1 | *-30.5* |
| PaLM | $I^{GenArg}$ | 58.1 | 46.6 | *-11.5* | 59.9 | 47.3 | *-12.6* | 78.1 | 84.4 | *+6.3* | 85.4 | 78.7 | *-6.7* |

Table 5: LLM performance on subsets where $\Lambda/\Phi$ is *consistent/adversarial* to entailment labels, measured with $AUC_{norm}$ (0% = random chance performance). Decrease from *cons* to *adv* subsets are shown in the *diff.* columns.

subsets w.r.t. $\Phi$ is widened from 10.1% to 16.1% for Levy/Holt and from 14.8% to 16.5% for RTE-1. The differences for $\Phi$-consistency subsets are generally narrower than $\Lambda$-consistency subsets, possibly because the relative frequencies require generalizing from instances, and may be more difficult to capture, and potentially because frequency measures with Google N-gram are a crude estimate of the actual frequencies in LLM pre-train corpora.

## 9  Conclusion

Across several major LLM families and experimental settings, we demonstrate two important biases in the performance of LLMs on natural language inference tasks, which may also manifest in applied tasks as hallucination. Contrary to claims of LLM general reasoning capabilities, we show that much of this performance is achieved by (1) recall of relevant memorizations and (2) corpus-based biases like term frequency. Since these factors are reproduced in all models, we establish that they originate in LLM pre-training, and are not corrected during GPT-3.5 fine-tuning.

We conclude that LLMs, though powerful, use unsatisfactory tools for the basic tasks of language understanding and inference. We propose several approaches to control for these biases in evaluation, and ultimately conclude that further attention on alleviating these biases are needed, before LLMs may be trusted to reason robustly about language.

## Limitations

In this paper, we have discussed two prominent sources of hallucination for LLMs in natural language inference tasks. We acknowledge that this is not an exhaustive search of all the sources, where further exploration should be done in future work.

We also note that after controlling for the factors discussed in this paper, there remains residual, unexplained performance on NLI tasks. This residual might be due to other undiscovered biases or possibly generalising inference capability. We leave further exploration of this residual to future work.

As discussed in Appendix A, we compared a range of popular LLM prompting techniques and selected the most promising approach. We acknowledge that there could potentially be other novel prompting techniques that could help the LLMs resist the influence of the biases discussed in this paper. We identify this as an open question and advocate for future research.

## Ethical Considerations

This paper discusses two major sources of hallucination in LLM output when asked to perform natural language inference, which we note is a capability required of many downstream tasks such as summarization, question answering, etc. We show that users of LLMs may be subjected to faulty judgements if the content of their request overlaps with data in pretraining. However, it is difficult to ascertain for both a user or modeler exactly what is contained in pretraining data, or how this will interact with a user's query. Our proposed attestation query shows promise in detecting potential overlaps, but model responses in applications of these cases are not explored. Further, the relative frequency bias demonstrates a much more subtle problem of corpus distribution that is naturally inherent to model pretraining on human generated text.

In light of these, the potential harms of LLM use for drawing natural language inferences may include: offering inaccurate or irrelevant information to a user's query or contradiction of information provided in-context with a user's query.

## Acknowledgements

This research was supported by ERC Advanced Fellowship GA 742137 SEMANTAX and the University of Edinburgh Huawei Laboratory.

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

## A Prompt Format Selection

In prompt-based interactions with the LLMs, several types of context information could be added to help models produce accurate and robust predictions. We attend to two design choices in prompt engineering: prompt templates and in-context examples.

**Prompt templates** are known to have a direct and sometimes decisive impact on LLM behavior. As such, we carefully select a range of clear and concise templates as promising candidates. As discussed in §4.2, we run each template through the dev sets of each dataset, and select the template with the best discriminative power according to AUC scores (similarly to §8). The candidate set of templates includes 3 concise templates we wrote:

1. If [PREMISE], then [HYPOTHESIS].

2. [PREMISE], so [HYPOTHESIS].

3. [PREMISE] entails [HYPOTHESIS].

We also considered the 5 prompt templates used in bias work on LMs for textual entailments (Schmitt and Schütze, 2021):

4. [PREMISE], which means that [HYPOTHESIS].

5. [HYPOTHESIS], because [PREMISE].

6. It is not the case that [HYPOTHESIS], let alone that [PREMISE].

7. $[\text{HYPOTHESIS}]_{NEG}$, which means that $[\text{PREMISE}]_{NEG}$.

8. $[\text{PREMISE}]_{NEG}$, because $[\text{HYPOTHESIS}]_{NEG}$.

In preliminary experiments with GPT-3.5, we observed that LLMs are not responsive to the 3 contrapositive prompts from Schmitt and Schütze (2021) (colored gray), performing at random. We also observed that prompt number 5 from Schmitt and Schütze (2021) also consistently underperforms the other 4 templates, so we use the remaining 4 templates (namely, template no. 1, 2, 3, 4) as our final candidate set.

**In-Context Examples** have been widely used for interactions with LLMs since Brown et al. (2020). Further, Wei et al. (2022) has demonstrated that including chain-of-thought explanation, namely step-by-step explanations, in the in-context examples, helps LLMs perform reasoning tasks. On the other hand, Ouyang et al. (2022) has suggested that instruction-tuned LLMs are also capable of performing tasks in zero-shot, without exposure to any in-context examples.

We compared zero-shot and few-shot in our preliminary experiments with LLaMA and GPT-3.5 on Levy/Holt directional **dev** set. Following Touvron et al. (2023), for zero-shot, we prepend a textual description of the task to each test sample; for few-shot, we prepend a minimal 4 examples with explanations. Instantiated prompts in the two settings are demonstrated in Table 13. Here we report the dev set results with the best-performing templates.

We found that for the two pre-trained LLMs, namely, LLaMA and PaLM, zero-shot performance on the Levy/Holt directional dev set is near-random, at 56.6% and 61.5% $AUC$ respectively (random is 50%); with 4 in-context examples, the models begin to exhibit non-trivial behavior, with 65.0% and 80.2% $AUC$, respectively. This is not surprising, since pre-trained LLMs without instruction fine-tuning should not be expected to perform complex tasks zero-shot. For GPT-3.5, the performance is still much lower in zero-shot, at 64.5%, compared to 74.6% in few-shot.

As discussed in §4.2, ideally we would like LLMs to have zero-shot natural language abilities readily available for downstream tasks. However, in light of this observation, our primary experiments are conducted in the few-shot setting

| Model | Task | Precision | Recall | $\Delta$-*Recall* |
|-------|------|-----------|--------|-------------------|
| LLaMA | $I$ | 74.5 | 52.5 | *0* |
| | $I^{GenArg}$ | 70.9 | 57.3 | *+4.8* |
| | $I^{RandArg\downarrow}$ | 66.9 | **60.5** | *+8.0* |
| | $I^{RandArg\uparrow}$ | 70.6 | 51.5 | *-1.0* |
| GPT-3.5 | $I$ | 80.6 | **96.5** | *0* |
| | $I^{GenArg}$ | 79.7 | 91.3 | *-5.2* |
| | $I^{RandArg\downarrow}$ | 80.1 | 82.5 | *-14.0* |
| | $I^{RandArg\uparrow}$ | 81.9 | 80.5 | *-16.0* |
| PaLM | $I$ | 90.3 | **84.0** | *0* |
| | $I^{GenArg}$ | 92.3 | 71.5 | *-12.5* |
| | $I^{RandArg\downarrow}$ | 87.8 | 82.5 | *-1.5* |
| | $I^{RandArg\uparrow}$ | 88.2 | 82.0 | *-2.0* |

Table 6: Scoring model outputs in different conditions of RTE-1. We indicate the **highest** and lowest recall score across replacement settings.

throughout, in order to better explore the abilities of these LLMs.

## B   RTE-1 Results For Experiment 2: Entities are Indices to Memory

The RTE-1 dataset contains complex natural language statements with varied linguistic features, so predictions about entailment are not decidable only on the basis of contained predicates. However, RTE-1 is a difficult challenge set for models, and interesting to compare to in the broader domain of NLI. Though the sentences are much more complex, we are able to conduct an analogous experiment as in §6 by first identifying spans of named entities and their respective entity types, then replacing the entities with new ones. As before, we compare model scores on the original dataset to three test conditions: generic arguments ("location X", "person Y", etc.), sampled low-frequency entities constrained to the same type, and the same for high-frequency sampled entities. Since only the entities in each statement have been altered, the entailment labels between premise/hypothesis pairs remain unchanged, and an ideal model capable of generalizing inference would make the same predictions across dataset conditions. Results are shown in Table 6.

We observe similar trends to those reported on Levy/Holt. GPT-3.5 performs very consistently between Levy/Holt and RTE-1 in terms of degrading recall when information is changed in each sample. We observe that model performance is worse than the original dataset when using generic arguments, and worse still using type-constrained random ar-

guments. We further observe that across all three LLMs across both datasets, models consistently achieve worse recall using high-frequency entities than low-frequency entities, supporting the claim that increasing the frequency of entity occurrence in training data impedes generalization.

Different from in Levy/Holt, we observe some noise in LLaMA's predictions; the recall on the original task is actually lower than the generic argument condition and the low-frequency entity condition. We note that overall, LLaMA is the weakest LLM tested in this experiment on both Levy/Holt and RTE-1, and that its performance on RTE-1 is particularly low. We suggest that the increased difficulty of RTE-1 over Levy/Holt (due to having much more linguistic variation) is simply too complex for LLaMA, which is neither the largest LLM tested, nor instruction-finetuned.

We also observe a smaller gap between PaLM's recall rates across dataset conditions, though the gaps are consistent with our claims. While the model appears able to generalize to conditions in which random real arguments are inserted, recall on the generic argument condition is significantly degraded. Failure on this control condition indicates that the model may not be generalizing as well as the other conditions would imply.

## C   The Ineffectiveness of Instructing LLMs to Stop Conditioning on Attested Information

In §5 and §6, we showed that entailment predictions from LLMs are strongly biased by their predictions on the attestation of hypotheses. We wondered whether there are intuitive prompt engineering techniques to steer its behavior away from attending to attestation.

Towards this goal, we experimented with prepending a brief task description to the few-shot prompts in part B of Table 13, explicitly instructing the models to ignore the attestedness of individual statements: *Please check the entailments between the following hypothetical statements. Ignore the veracity of these statements.*

We replicated the experiments in §5 and §6 with GPT-3.5, since GPT-3.5 is an instruction-finetuned model trained to be responsive to prompts, where the other two LLM families are only pre-trained. Despite having been instruction-finetuned, the results with GPT-3.5 show only marginal improvements in model behavior.

| task | GPT-3.5 | Instructed to Ignore Attestedness | Not Instructed |
|------|---------|-----------------------------------|----------------|
| $I$ | $P(\texttt{Entail} \mid \texttt{Attested})$ | 74.3 | 77.6 |
| $I$ | $P(\texttt{Entail} \mid \neg\texttt{Attested})$ | 57.8 | 63.6 |
| $I_{RandPrem}$ | $P(\texttt{Entail} \mid \texttt{Attested})$ | 39.0 | 41.3 |
| $I_{RandPrem}$ | $P(\texttt{Entail} \mid \neg\texttt{Attested})$ | 17.6 | 18.8 |

Table 7: We estimate the probability of positive predictions of `Entail` in $I$ and $I_{RandPrem}$ tasks respectively given that the hypothesis is attested, namely $\Lambda = \texttt{Attested}$. **Not instructed** results are copied from Figure 2 and listed here for ease of comparison; also note that all $I_{RandPrem} = \texttt{Entail}$ predictions are false positives.

| GPT-3.5 Condition | Task | Levy/Holt (Directional) | | |
|-------------------|------|-----------|--------|----------|
| | | Precision | Recall | $\Delta$-**Recall** |
| | $I$ | 64.9 | **90.8** | 0 |
| Few-shot, instructed to ignore attestedness. | $I^{GenArg}$ | 73.5 | 69.3 | -21.5 |
| | $I^{RandArg\downarrow}$ | 64.6 | 68.4 | -22.4 |
| | $I^{RandArg\uparrow}$ | 67.5 | 58.1 | -32.7 |
| | $I$ | 62.4 | **92.3** | 0 |
| Few-shot, no instructions. | $I^{GenArg}$ | 65.1 | 75.7 | -16.6 |
| | $I^{RandArg\downarrow}$ | 65.5 | 66.5 | -25.8 |
| | $I^{RandArg\uparrow}$ | 68.8 | 55.3 | -37.0 |

Table 8: GPT-3.5 predictions when models are explicitly instructed to avoid taking the attestedness of individual statements into account. In the upper half are the instructed behavior, and in the lower half are the regular few-shot behavior as in Table 4. Differences in recalls remain at a similar scale, with precision again stable, where the benefit from the explicit instruction is marginal.

In Table 7, we show that instructing GPT-3.5 to ignore attestation does not help narrow the gap between $\Lambda = \texttt{Attested}$ and $\Lambda = \neg\texttt{Attested}$; instead, probabilities of predicting `Entail` went down by similar amounts, indicating that the model is becoming slightly more conservative in predicting positives when instructed to ignore attestation, but not in a principled manner.

Further, as shown in Table 8, despite the explicit instruction, recall still drops at similar scales when arguments are randomly replaced with the same sets of frequent/infrequent replacement entities as before. Since GPT-3.5 has been instruction-finetuned to respond to prompts, its failure means eradicating such biases from model outputs is a difficult task, one that needs further research attention.

## D  Statistics of Consistency Subsets

The statistics of consistency subsets are presented in Table 9.

## E  The Reliability of $\Lambda$ Measure and Its Relation to Consensus of Attestation

The $\Lambda$-consistency subsets most directly capture the impacts of the *attestation bias*. However, these subset separations are based on $\Lambda$ predictions from individual models, which can be noisy, subject to model-specific idiosyncrasies such as trigger words or certain syntactic structures in the prompt, etc.

To verify that the performance gaps in $\Lambda$-consistency subsets that we observe in §8 comes from predicted attestedness and not some idiosyncrasy, we experiment with another pair of subsets based on *consensus attestation* instead of individually *predicted attestation*.

We use a majority vote among the three independently-trained LLMs to approximate *consensus attestation*. The approximation is denoted as $\tilde{\Lambda}$. This is because any model-specific idiosyncrasies should not be shared between LLMs independently trained from different source corpora in general. Therefore, with the majority vote, we reduce this noise and acquire predictions on the *consensus attestation* of statements.

Performances of LLMs between $\tilde{\Lambda}$-consistency subsets are listed in Table 10. Gaps between the $\tilde{\Lambda}$-consistency subsets that are larger than $\Lambda$-consistency gaps are colored red; those narrower than $\Lambda$-consistency gaps are colored green. It is clear that the gaps are consistent between $\Lambda/\tilde{\Lambda}$-consistency experiments, where the gaps are even larger on many occasions. This confirms that the

| # of Entries | Levy/Holt | | | RTE-1 | | |
|---|---|---|---|---|---|---|
| | LLaMA | GPT-3.5 | PaLM | LLaMA | GPT-3.5 | PaLM |
| $V_{\text{CONSISTENT}}$ | 955 | 947 | 999 | 479 | 447 | 480 |
| $V_{\text{ADVERSARIAL}}$ | 829 | 837 | 785 | 321 | 353 | 320 |
| $F_{\text{CONSISTENT}}$ | | 972 | | | 286 | |
| $F_{\text{ADVERSARIAL}}$ | | 220 | | | 247 | |

Table 9: Subsets defined by the consistency between entailment label $L$ and either $\Lambda$ (hypothesis attestation prediction from each LLM) or $\Phi$ (model-agnostic relative frequency bias). CONSISTENT subsets are where $L$ agrees with $\Lambda/\Phi$. ADVERSARIAL subsets are where $L$ disagrees with $\Lambda/\Phi$.

| | | Levy/Holt | | |
|---|---|---|---|---|
| Model | Task | $\tilde{\Lambda}_{cons.}$ | $\tilde{\Lambda}_{adv.}$ | diff. |
| LLaMA | $I$ | 65.3 | 6.5 | *-58.8* |
| GPT-3.5 | $I$ | 70.8 | 23.5 | *-47.3* |
| PaLM | $I$ | 80.7 | 28.3 | *-52.4* |
| LLaMA | $I^{GenArg}$ | 54.4 | 29.6 | *-24.8* |
| GPT-3.5 | $I^{GenArg}$ | 56.2 | 35.5 | *-20.7* |
| PaLM | $I^{GenArg}$ | 59.3 | 40.1 | *-19.2* |

Table 10: LLM performance on Levy/Holt subsets where Attestation $\tilde{\Lambda}$ is Consistent/Adversarial to the labels, measured with $AUC_{norm}$ (0% = random chance performance). Performance drops from $\tilde{\Lambda}_{cons}$ to $\tilde{\Lambda}_{adv}$ are presented in the *diff.* columns, sharper decreases than $\Lambda$-comparisons in Table 5 are colored **red**, milder ones are colored **green**.

performance gaps in $\Lambda$-consistency experiments can be credited to the *attestation bias*, rather than model-specific idiosyncrasies.

It is also to be noted that, since the $\Phi$-consistency subsets are separated based on the model-agnostic criterion $\Phi$, model-specific idiosyncrasies are not a problem for $\Phi$-consistency comparisons.

## F  Impacts of Bias on GPT-4 Performance

GPT-4 (OpenAI, 2023) is a recent, strong LLM claiming SOTA performance on various NLP tasks. Due to its closed-source nature and the impossibility of fully tracking the sources of its behaviors, we refrain from reporting results with it in the main content of this paper.

However, in order to provide a richer context for the *attestation bias* and the *relative frequency bias*, in this section we report the performance differences of GPT-4 between subsets consistent/adversarial to the two biases.

As a light-weight experiment, we elicit GPT-4 predictions in the original $I$ task in the zero-shot setting, and re-use subsets from experiments in §8. Specifically, for the *attestation bias*, we use

the majority vote $\tilde{\Lambda}$ among LLaMA, GPT-3.5 and PaLM, to approximate $\Lambda$ predictions from GPT-4 itself; for the *relative frequency bias*, we keep the $\Phi$ measure for approximating corpus-frequency of terms.

Because GPT-4 is a commercial service and does not provide logit confidence with their discrete predictions, $AUC_{norm}$ values could not be calculated. Therefore, we are forced to report **the *F-1 scores* at the binary prediction point of confidence**. As results in Table 12 show, we observe the same trend as in §8: for the subset adversarial to each factor, GPT-4 performance also drops substantially.

This experiment is designed to provide more context for the two biases discussed in the paper and **NOT** to compare GPT-4 with other models; however, we can conclude that GPT-4 is subject to the same fragilities as the other LLMs w.r.t. the two biases, where our conclusions and recommendations also apply.

## G  Dataset Statistics and Dev Set Performance

In the paper, we have examined the behavior and performance of three major LLM families on two NLI datasets: Levy/Holt and RTE-1.

The directional portion of Levy/Holt dataset[9] contains 630 entries in its dev set, and 1784 entries in its test set; the RTE-1 dataset[10] contains 567 entries in its dev set, and 800 entries in its test set. Each dataset has a 50%/50% class distribution between `Entail` and `No-Entail` (for RTE-1 dev set, the numbers of entries in the two label classes differ by 1).

In Table 11, we report dev set performance and the best prompt template used for each model on each dataset. Note that no training is involved in

---

[9] https://github.com/mjhosseini/entgraph_eval/tree/master/LevyHoltDS
[10] https://www.kaggle.com/datasets/nltkdata/rte-corpus?resource=download

| Model | Task | Levy/Holt | | RTE-1 | |
|---|---|---|---|---|---|
| | | Best tplt. ID | DEV set $AUC_{norm}$ | Best tplt. ID | DEV set $AUC_{norm}$ |
| LLaMA | $I$ | #4 | 30.0 | #3 | 62.5 |
| | $I^{GenArg}$ | #1 | 34.6 | #3 | 52.3 |
| | $I^{RandArg\downarrow}$ | #1 | 31.8 | #1 | 51.3 |
| | $I^{RandArg\uparrow}$ | #1 | 26.3 | #3 | 43.8 |
| GPT-3.5 | $I$ | #1 | 49.2 | #3 | 74.8 |
| | $I^{GenArg}$ | #1 | 39.8 | #3 | 64.8 |
| | $I^{RandArg\downarrow}$ | #1 | 43.4 | #3 | 63.6 |
| | $I^{RandArg\uparrow}$ | #1 | 34.2 | #3 | 66.0 |
| PaLM | $I$ | #1 | 60.9 | #4 | 84.5 |
| | $I^{GenArg}$ | #1 | 48.1 | #4 | 79.4 |
| | $I^{RandArg\downarrow}$ | #1 | 43.6 | #3 | 79.8 |
| | $I^{RandArg\uparrow}$ | #1 | 35.3 | #3 | 78.3 |

Table 11: LLM **dev set** performance on the two datasets, measured with $AUC_{norm}$ (0% = random chance performance). AUC is calculated using estimated model scores as in §4.2 and then normalized into AUC$_{norm}$. We select the highest scoring template on each dev task (shown in this table) and use this in the corresponding test set evaluation (shown in the main text).

| F-1 score | Task | Levy/Holt | |
|---|---|---|---|
| | | $\tilde{\Lambda}_{Cons}$ | $\tilde{\Lambda}_{Adv}$ |
| *random baseline* | $I$ | 70.3 | 62.0 |
| GPT-4 | $I$ | 85.1 (**+14.8**) | 67.6 (+5.6) |
| | | $\Phi_{Cons}$ | $\Phi_{Adv}$ |
| *random baseline* | $I$ | 66.7 | 66.7 |
| GPT-4 | $I$ | 74.6 (**+7.9**) | 69.7 (+3.0) |

Table 12: LLM performance on Levy/Holt subsets where Attestation $\tilde{\Lambda}$ is Consistent/Adversarial to the labels, measured with **F-1 score**. *random baseline* is the highest F-1 score from a random classifier, by reaching random precision and 100% recall. For each GPT-4 score, we also show the improvement over random (in parentheses).

this paper, and prompt template selection is the only hyper-parameter tuned on the dev sets. These selected best prompt templates are then used on the respective test sets, where the results are used for the analysis throughout the paper.

For random-premise experiments, AUC values cannot be meaningfully calculated because gold labels are always `No-Entail`. For these experiments, we use the most frequently-selected prompt template on each dataset, namely template #1 for Levy/Holt dataset, and template #3 for RTE-1 dataset.

---

**A. Zero-shot Example Instantiated Prompt**

---

Please check the entailments between the following statements.

If kanamycin kills infections, then kanamycin is useful in infections.
A) Entailment
B) Neutral
C) Contradiction

---

**B. Few-shot Example Instantiated Prompt**

---

If Google bought Youtube, then Google owns Youtube.
A) Entailment
B) Neutral
C) Contradiction
Answer: A) Entailment. Owning is a consequence of buying.
If Google owns Youtube, then Google bought Youtube.
A) Entailment
B) Neutral
C) Contradiction
Answer: B) Neutral. Owning does not imply buying, the ownership may come from other means.
If John went to the mall, then John drove to the mall.
A) Entailment
B) Neutral
C) Contradiction
Answer: B) Neutral. John may have gone to the mall by other means.
If John drove to the mall, then John went to the mall.
A) Entailment
B) Neutral
C) Contradiction
Answer: A) Entailment. Driving is a means of going to the mall.
If ephedrine is widely used in medicine, then ephedrine is used in medicine.
A) Entailment
B) Neutral
C) Contradiction
Answer:

---

**C. Hypothesis-only Example Instantiated Prompt**

---

Google bought Youtube.
A) True
B) Unknown
C) False
Answer: A) True.
Yoshua Bengio likes oak trees.
A) True
B) Unknown
C) False
Answer: B) Unknown.
The sun rises from the west.
A) True
B) Unknown
C) False
Answer: C) False.
ephedrine is used in medicine.
A) True
B) Unknown
C) False
Answer:

---

Table 13: Example instantiated prompts in Zero-shot / Few-shot settings, for the sample "PREMISE: [ephedrine is widely used in medicine], HYPOTHESIS: [ephedrine is used in medicine]". The few-shot prompts in part B are used throughout the main experiments in this paper. We also present an example of the prompts we use for the hypothesis-only $\Lambda$ measure as described in §3.1.