# OpenReview forum: "Sources of Hallucination by Large Language Models on Inference Tasks"
_EMNLP/2023/Conference — EMNLP 2023 Findings_

### Official Review · Reviewer_oXrA · 2023-08-03

**Soundness:** 3

**Excitement:**

3: Ambivalent: It has merits (e.g., it reports state-of-the-art results, the idea is nice), but there are key weaknesses (e.g., it describes incremental work), and it can significantly benefit from another round of revision. However, I won't object to accepting it if my co-reviewers champion it.

**Paper Topic And Main Contributions:**

The paper addresses the phenomenon of hallucinations in LLMs when performing inference tasks. The paper specifically focuses on cases when LLMs wrongfully predicts entailment. The authors suggest that the origin of such hallucinations stems from two factors: the presence of the hypothesis in the pretraining data (the attestation bias),  and the statistical patterns of the pretraining data (the relative frequency bias), which the authors demonstrate in their experiments.

The paper’s main contribution is the proposed source of inference hallucinations, which could lead to better training and deployment of LLMs, in order to mitigate this behavior


**Reasons To Accept:**

1.	The theory proposed in this paper is of important value in the task of characterization and mitigation of hallucinatory behavior in LLMs.

**Reasons To Reject:**

1.	It was quite unclear how the experiments performed in the work to corroborate the authors’ theory did that. In the random premise task – how could the authors ensure that the random predicate indeed resulted in NO-ENTAIL? I understand that such random sampling has a very small probability of resulting in something that is not NO-ENTAIL, but given that many predicates have synonyms, and other predicates whom they entail, and hence by proxy the current hypothesis might also entail, it feels like it is crucial to ensure that indeed NO-ENTAIL was the case for all the instances (as this is not the train set, but rather the evaluation set).
2.	Additionally, it was not clear how the generic argument task and the random argument task proved what the authors claimed. All in all, the whole dataset transformation and the ensuing experimental setup felt very cumbersome, and not very clear.


**Reproducibility:**

4: Could mostly reproduce the results, but there may be some variation because of sample variance or minor variations in their interpretation of the protocol or method.

**Reviewer Confidence:**

4: Quite sure. I tried to check the important points carefully. It's unlikely, though conceivable, that I missed something that should affect my ratings.

---

> ### Author Rebuttal · Authors · 2023-08-28
>
> Thank you for your time and thoughtful comments on the paper! We’d like to respond to your 2 concerns in “Reasons to Reject”:
>
> #### **Question 1: How do we ensure No-Entail labels in the Random Premise Experiment?**
>
> In initial experiments, manual inspection of a sample of random-premise generations looked acceptable to the authors. But this is a valid concern, and we have since conducted an additional manual expert annotation over the entire set of 1784 random-premise entries to filter out any generated ENTAIL samples. Our new manual expert annotation identified 171 ENTAIL entries (9.8% of generated entries) and confirmed 1545 NO-ENTAIL entries. The remaining 68 are marginal cases which could be interpreted in either direction depending on the reading. With the limited time-frame in mind, we simply removed the 171+68=239 entries of ENTAIL / marginal ENTAIL from the dataset, and recalculated the probabilities.
>
> **The revised probabilities leave the conclusions of the paper unchanged.** If accepted, we will add them in an appendix. Specifically:
>
> 1. First, we note that the main paper claim is observed strongly in the original dataset itself (Figure 1): all 3 LLM families affirm inference questions more often when the hypothesis is attested by training data. The random-premise transformation is offered mainly as a further control (L398), on which the models continue to make many positive judgments, showing that they are answering questions using hypothesis attestation, instead of considering if the (now changed) premise entails it.
>
> 2. With the new cleaned data subset, we re-estimated the probability values when conditioning on the Attestation bias (Figure 2), which we present here in the same left-to-right order. We note that the effect size is very strong (the gaps are wide between attested/non-attested probabilities of predicting ENTAIL).
>
> | Estimated Probability        | LLaMA       | GPT-3.5     | Anon. LM   |
> | ---------------------------------- | -------------- | -------------- | --------------- |
> | Before Cleaning (1784)  &emsp;   | 39.7 / 20.7 &emsp;| 41.3 / 18.8 &emsp; | 39.9 / 19.9 |
> | Clean NO-ENTAIL (1545) | 35.9 / 16.9 | 34.3 / 11.7 | 33.0 / 14.8 |
>
> 3. We also re-estimate the probabilities for conditioning on the Frequency Bias (Figure 3). The effect size is smaller, but it is consistent with all other results. We note that among the 239 entries removed for possible ENTAIL, many of their hypothesis predicates are high-frequency/semantically general, since many randomly selected premises will entail something generalized. E.g. 41 entries with hypothesis `lives in`, 14 `contains`, 8 `traveled to`, 7 `is in`, etc. Removing these leads to a distribution shift which especially affects the creation of subsets defined by the Relative Frequency bias conditioning (so results are less comparable before/after cleaning). We report the results (with the same left-to-right order as Figure 3).
>
> | Estimated Probability        | LLaMA                           | GPT-3.5                        | Anon. LM        |
> | ---------------------------------- | ------------------------------- | ------------------------------ | ------------------- |
> | Before Cleaning (1784) &emsp;    | (32.4 / 28.5), (28.2 / 17.9) &emsp; | (42.6 / 31.0), (24.6 / 14.0) &emsp; | (31.9 / 25.0), (11.8 / 5.8) |
> | Clean NO-ENTAIL (1545) | (27.7 / 26.4), (18.6 / 16.6) | (30.7 / 26.7), (14.1 / 11.5) | (25.3 / 20.1), (5.4 / 4.4) |
>
> #### **Question 2: How does Argument Replacement Show Memorization in Models?**
>
> LLM performance using original named entities degrades when we switch to the generic- and random-entity datasets, which demonstrates LLMs’ reliance on memory about the original entities (which tend to be “expected” in context), and LLMs’ inability to generalize about valid but unexpected entities. These transformations create samples where the predicates/entailment label remain the same as the original dataset, but each individual proposition becomes surprising to the LLMs.
>
> In the generic argument task, each individual proposition is meant to have neutral attestation, since “true” or “false” labels can rarely be assigned to claims about generic entity types (e.g. `PERSON resided in LOCATION` is neither true nor false). In the random argument task, each individual proposition may have neutral or negative attestation (e.g. `Helsinki exports Granny Smith Apples` is very unexpected and likely “false”/unattested by training data). With more frequent pairs of entities, the LLMs have a weaker ability to reason about novel information, because the mentions of these entities that are read during pre-training interfere. Therefore, we can generally observe a decreasing trend in recall from GenArg to RandArg⬇️ to RandArg⬆️, especially between RandArg⬇️ and RandArg⬆️, as shown in Table 3.

---

### Official Review · Reviewer_4oz7 · 2023-08-03

**Soundness:** 4

**Excitement:**

4: Strong: This paper deepens the understanding of some phenomenon or lowers the barriers to an existing research direction.

**Paper Topic And Main Contributions:**

The paper studies the origins of hallucination and impaired performance in the NLI task. It identified two heuristics learned by pretrained models: (1) they tend to predict entailment when the hypothesis (alone) it attested in the training data (2) they tend to predict entailment when the premise is less frequent in the training data than the hypothesis, reflecting a (non-causal) tendency in NLI data to infer the more abstract from the more concrete. Further, it is shown that it is the entities that serve as anchors in memorisation. Overall, this paper sheds light on important issues behind the apparent good performance of pretrained LMs on NLI data.

**Questions For The Authors:**

What is the relation between the hypothesis-attestment heuristic and the already well known hypothesis only baseline? do you see these findings as exposing the mechanism behind the hypothesis only baseline?

**Reasons To Accept:**

- The paper shows an important and easy-to-miss limitation of current LMs, highlighting the need to disentangle performance on specific datasets form performance on the general task of interest.

- Very elegant experimental design, both showing evidence for the two new heuristics, and isolating the mechanism by which the model memorizes.


**Reasons To Reject:**

I don't see any reason to reject the paper.

One technical limitation is the use of proxies for occurrence in the training data. Hopefully indexing the pretraining data of large LMs would enable measuring to what extent these proxies actually reflect occurrence in the training data.

**Reproducibility:**

4: Could mostly reproduce the results, but there may be some variation because of sample variance or minor variations in their interpretation of the protocol or method.

**Reviewer Confidence:**

3: Pretty sure, but there's a chance I missed something. Although I have a good feel for this area in general, I did not carefully check the paper's details, e.g., the math, experimental design, or novelty.

---

> ### Author Rebuttal · Authors · 2023-08-28
>
> Thank you for your time and thoughtful analysis of the paper! To answer the question:
>
> The hypothesis-only baseline of Poliak et al (2018) trains a model to make judgements on NLI datasets based only on the hypothesis, without considering the premise. Through training it is able to detect statistical artifacts like word choice and grammaticality and make predictions much better than chance.
>
> Our hypothesis-attestation judgements are obtained without any fine-tuning, simply by asking the model whether the hypothesis statement is true, false, or uncertain. The models perform better on samples which align with this bias (Entail when hypothesis is judged true, No-Entail when hypothesis is not), than on samples that do not.
>
> Between the two lines of work, the hypothesis-only baseline of Poliak at al focuses on detecting biases in NLI training datasets which could be exploited by models, when trained on these datasets; on the other hand, our hypothesis-attestation conditioning indicates that the class of LLMs studied is inherently sensitive to attestation of hypotheses, separate from the idiosyncrasies of particular datasets.
>
> However, we believe it may also be possible that the models tested in Poliak et al, being BERT-based models, could also be subject to the same attestation bias as observed with larger LMs in this paper. This attestation bias could be contributing to their hypothesis-only performance in addition to the explicit dataset artifacts which are exposed in their paper. It would be interesting to test for this in future work using an attestation control on those models and datasets.

---

### Official Review · Reviewer_Fdep · 2023-08-05

**Soundness:** 3

**Excitement:**

4: Strong: This paper deepens the understanding of some phenomenon or lowers the barriers to an existing research direction.

**Paper Topic And Main Contributions:**

This paper presents a behavioral study of 3 LLMS which shows that some biases are major sources of hallucination in generative LLMs. They define two biases for which they modify two existing NLI datasets in a way in which they can control the bias during generating the correct label for NLI instances.

**Reasons To Accept:**

Overall I think the the paper presents a nice and creative methodology for showing that biases may lead LLMs to hallucinate. At the beginning I had a hard time understanding the main contribution of the paper; however, once understood, it believe the conclusions made by this work are significant to the NLP field.

**Reasons To Reject:**

While the contribution may be considered as significant, I believe that it might be a bit limited (only NLI, only 3 LLMs) to be considered for the main EMNLP stage. Also I find the paper a bit hard to follow. There are not enough examples, and it took me a lot of time to understand the methodology while I believe it could be described in a better way (for example, relative frequency bias should be better explained). Other than that I have no major concerns.

**Reproducibility:**

4: Could mostly reproduce the results, but there may be some variation because of sample variance or minor variations in their interpretation of the protocol or method.

**Reviewer Confidence:**

3: Pretty sure, but there's a chance I missed something. Although I have a good feel for this area in general, I did not carefully check the paper's details, e.g., the math, experimental design, or novelty.

**Typos Grammar Style And Presentation Improvements:**

It is hard to understand directly from the abstract what is the main contribution of the paper. I suggest adding the bottom line qualifications to the abstract to make it clearer.

---

> ### Author Rebuttal · Authors · 2023-08-28
>
> Thank you for your time and thoughtful comments on the paper!
>
> Regarding the comment about limited contribution, we note that we have experimented with 3 different LLM families (trained independently), on 2 datasets, with 8 prompt templates. We argue this is sufficiently representative of the current generation of LLMs on the NLI task. For the anonymous LLM, although we are unable to reveal its identity until the end of the anonymity period, we are able to confirm that it is a SOTA LLM of a larger scale than GPT-3. We focus on the task of textual inference because it is fundamental to the many NLP tasks that require more than mere approximation to memorized training data. Examples are: retrieval-aided question-answering from documents that entail but do not state a direct answer to the question; or multi-document summarization, where redundancy of statements in one document that are entailed by others must be detected.
>
> We agree that adding more illustrative examples is a great idea, and can help explain the relative frequency bias. We already have examples for all experimental conditions in Tables 1 and 2, but we believe that adding a dev set example which illustrates when each bias aligns with the correct label will help introduce them much better in S3.1. We will add these:
>
> ---
>
> **Attestation:**
>
> Prem: Amon became the god of Egypt
>
> Hyp: Amon was worshiped in Egypt
>
> Label: Entail
>
> Hyp attested? Yes
>
> ---
>
> Prem: Amon was worshiped in Egypt
>
> Hyp: Amon became the god of Egypt
>
> Label: No-Entail
>
> Hyp attested? No (Egypt was polytheistic in the time of Amon)
>
> ---
>
> **Frequency:**
>
> Prem: Whiskey consists chiefly of alcohol
>
> Hyp: Whiskey contains alcohol
>
> Label: Entail
>
> freq(“consists chiefly of”) < freq(“contains”)? Yes
>
> ---
>
> Prem: Whiskey contains alcohol
>
> Hyp: Whiskey consists chiefly of alcohol
>
> Label: No-Entail
>
> freq(“contains”) < freq(“consists chiefly of”)? No
>
> ---
>
> We also plan to further clarify the introduction to the relative frequency bias by elaborating on the existing reference to McKenna (2022). Very infrequent predicates tend to be very specific (e.g. “perambulate,” “hike”) compared to very frequent predicates which tend to be more semantically general (e.g. “walk,” “move”). A specific predicate may entail a general one (e.g. “hike” entails “walk”) but the reverse is not possible (“walk” does NOT entail “hike”). Language Models are known to be sensitive to frequency, and we show this accounts for some capability to detect entailment.

---

### Meta-Review · Area_Chair_7Fnq · 2023-09-26

**Recommendation:** 3

**Metareview:**

The paper establishes two sources of hallucinations in LLMs: memorization and statistical patterns. While reviewers see merit in the work, they also express reservation about whether the conclusions are sufficiently supported.

---

### Decision · Program_Chairs · 2023-10-07

**Decision:**

Accept-Findings

**Comment:**

The paper establishes two sources of hallucinations in LLMs: memorization and statistical patterns. While reviewers see merit in the work, they also express reservation about whether the conclusions are sufficiently supported.